# Occupational exposure to blood/body fluid splash and its predictors among midwives working in public health institutions at Addis Ababa city Ethiopia, 2020. Institution-based cross-sectional study

**Solomon Shitu** [1]*, **Getachew Adugna**[2], **Haimanot Abebe** [3]

**1** Department of Midwifery, Wolkite University College of Health and Medical Sciences, Wolkite University, Wolkite, Ethiopia, **2** City Health Offices, Addis Ababa City Health Departments, Addis Ababa, Ethiopia, **3** Department of Public Health, Wolkite University College of Health and Medical Sciences, Wolkite University, Wolkite, Ethiopia

* solomonsht7@gmail.com

**Data Availability Statement:** All relevant data are within the manuscript and its Supporting Information files.

## Abstract

### Background

Blood/body fluid splash are hazards to health care professionals in their working area. Around twenty bloodborne pathogens are known to be transmitted through these occupational injuries. This problem alters the health status of health care professionals in different ways, including physically, mentally, and psychologically. Even though health professionals especially midwives who are working in delivery rooms are highly affected, little is known about the exposure. So, this study was aimed to assess the prevalence of exposure to blood/body fluid splash and its predictors among midwives working in public health institutions of Addis Ababa city.

### Methods

Institution based cross-sectional study was conducted among 438 study participants in public health institutions in Addis Ababa. Data was collected from March 1–20, 2020 by a self-administered questionnaire. The data were entered into Epi data version 3.1 and then exported to SPSS version 24 for analysis. All variables with P<0.25 in the bivariate analysis were included in a final model and statistical significance was declared at P< 0.05.

### Results

In this study, a total of 424 respondents respond yielding a response rate of 97%. The prevalence of blood and body fluid splashes (BBFs) was 198 (46.7%). Not training on infection prevention, working in two shifts (> 12 hours), not regularly apply universal precautions, job-related stress, an average monthly salary of 5001–8000 were independent predictors of blood and body fluid splashes.

**Funding:** The authors received no specific funding for this work

**Competing interests:** The authors have declared that no competing interests exist.

**Abbreviations:** AIDS, Acquired Immune Deficiency Syndrome; AOR, Adjusted Odds Ratio; BBFS, Blood/Body Fluid splash; CDC, Center for Disease Control; CI, Confidence Interval; COR, Crude Odds Ratio; FMoH, Federal Ministry of Health; HBV, Hepatitis B virus; HCV, Hepatitis C virus; HCPs, Health Care Providers; HIV, Human Immunodeficiency Virus; IP, Infection Prevention; OR, Odds Ratio; PEP, Post Exposure Prophylaxis; SPSS, Statistical Package for the Social Sciences; WHO, World Health Organization.

## Conclusion

The study revealed that nearly half of midwives were exposed to BBFS. This highlights the need for key stakeholders such as policymakers and service providers to design appropriate policies to avert this magnitude and making the environment enabling to comply with standard precautions. We recommend that this study may be done by including rural setting institutions and by including other health professionals that are susceptible to BBFS at work. Formal training on infection prevention and safety practice to apply universal precautions will be needed from the concerned bodies to prevent exposures to blood/body fluid splash.

## Introduction

Blood/body fluid splash (BBFS) are hazards to health care workers in their working area. Around twenty bloodborne pathogens are known to transmit through this occupational exposure [1]. This problem alters the health status of health care professionals (HCPs) in different ways, including physically, mentally, and psychologically [2].

No occupation or work is free from health hazards. People are always under a certain amount of health risk wherever they are working. Health care professionals are exposed to a wide range of occupational health and safety hazards. An estimated 20 million work-related injuries and 390, 000 new work-related illnesses occur each year in the United States and China 66.3% of HCPs per year exposed to BBFS; however, the number of occupational diseases and injuries reported each year is much more [3–5].

Nowadays, workplace injuries, especially BBFS prevalence, are increasing among health workers [6, 7]. Occupational exposure to BBFS in healthcare facilities constitutes a significant risk of transmission of human immune deficiency virus (HIV) and other blood-borne pathogens to healthcare workers [8–10].

As estimated by the Center for Disease Control (CDC) that every year more than three million HCPs are exposed to BBFs, through sharps, plus mucocutaneous injuries in the United States [10]. Due to BBFS, the possibility of infections ranges from as low as 0.2–0.5% for HIV to as huge as 3–10% for Hepatitis C virus (HCV) and 40% for Hepatitis B virus (HBV) [11]. World health organization (WHO) estimates that nearly 3 million HCPs experience occupational exposure to blood-borne viruses HBV, HCV, and HIV has been reported to be 2.1 million, 926,000, and 327,000 respectively [12–15]. From this 90% of this exposure to infection was resulted from low resource countries [14]. In Ethiopia, studies indicated that the prevalence of HCV due to BBFS exposure was 0.9–5.8% [3, 16]. Health care workers in Africa experience from two to four needle-stick injuries per year on average, with Nigeria, Tanzania, and South Africa reporting on average of 2.10 injuries per HCPs per year [17].

Even though there is no national data regarding the prevalence of BBFS among midwives in Ethiopia previous studies in the Eastern part of the country showed that the prevalence of BBFS was 28.8% [18]. The majority of these studies were only confined to a single institution and workplace [18–23]. A high rate of midwives is affected by the exposure [2]. Despite the presence of many possibilities to reduce the burden of occupational exposure to BBFS, it is not yet recognized. We included only midwives was as in general little is known about BBFS in all health professionals, and midwives' work is more focused on conducting delivery in which they are more susceptible for fluids and blood splash during attending delivery. Because they are nearest to exposure like amniotic fluid, meconium, blood, vomiting, and so on. Hence this research was carried out among midwives working at public health institutions of Addis Ababa city due to they are more exposed and the first line for BBFS.

## Methods

### Study design, area, and period

An institution-based cross-sectional study was conducted from March 1–20, 2020 in selected public health institutions of Addis Ababa Ethiopia's capital city. It comprises 10 sub-cities. In the City, there are 12 public hospitals and 112 health centers. All institutions serve the population of Addis Ababa and provide all services including maternal and child health care service [24, 25].

### Population

**Source population.** All midwives working in public health institutions of Addis Ababa city.

**Study population.** Randomly selected midwives working in public health institutions of Addis Ababa city.

### Inclusion criteria

All midwives working in Addis Ababa city.

### Exclusion criteria

Midwives who were on annual or maternity leave and who were critically ill and unable to respond during data collection.

### Sample size determination

Separate sample size was calculated for each specific objective by using both single and double population proportion formulas was used. For the first specific objective, (To determine the prevalence of occupational exposure to BBFS among midwives working in Addis Ababa city) single population proportion was used.

$n = \frac{(Z\alpha/2)^2 P(1-P)}{d^2}$ *by the* assumption of n = minimum sample size required for the study $(Z\,\alpha/2)^2$ = standard normal distribution with 95% confidence interval, P = 21% (prevalence of occupational exposure and behavior of health care workers in Ethiopia) [18], d = is a tolerable margin of error (d = 0.04).

The sample size of the first objective was greater than that of the second objective. By adding a 10% non-response rate from the largest sample size the calculated sample size was 438.

### Sampling procedure

Six public hospitals from 12 hospitals and 66 health centers from 112 health centers were selected by simple random sampling technique. The total number of midwives working in Addis Ababa city public health institutions was 2432 and the number of midwives working in selected public health institutions was 1296. To select study participants from selected health institutions the proportional allocation method (each institution's number of midwives was allocated by proportion) was used for each institution. Study participants who fulfilled the inclusion criteria were selected by systematic random sampling technique until the required number is obtained with a k[th] value of 3. The registers of all midwives were obtained from the administration of the offices. Any unwilling or unavailable selected participant was replaced by the very next person on the list who had not been already selected to participate.

### Dependent variables

Exposure to blood/body fluid splash.

### Independent variables

**Socio-demographic factors.** (Age, Sex, Marital status, Qualifications of midwives, Monthly income)

**Work-related factors.** (Training on IP, working environment, year of service, educational level)

**Behavioral factors.** (Job satisfaction, reporting after the exposure, using the universal precautions and personal protective equipment).

### Data quality control and collection methods

The questionnaire was developed by reviewing different pieces of literature [2, 3, 7, 17, 18, 26–29]. Data were collected by facilitator guided structured self-administered English version questionnaire. Facilitators took two days of training to assure quality. The questionnaire was checked for completeness before data entry into the software. Proper coding and categorization of data were maintained for the quality of the data to be analyzed. Double data entry was done for its validity and compare to the original data.

### Data processing and analysis

The collected data were checked and reviewed for completeness, coded, and entered into Epi data version 3.1, and exported to SPSS version 24 for analysis. Descriptive statistics were used to determine the frequency of different variables. Association between dependent and independent variables were examined using bi-variable and multivariable logistic regression models. Those variables which were (P-value < 0.25) [18] in the bivariate analysis were included in the final model of multivariate analysis to control all possible confounders. Multi-collinearity was checked to see the linear correlation among the independent variables by using standard error. Variables with a standard error of > 2 were dropped from the multivariable analysis. Model fitness was checked with the Hosmer-Lemeshow test. The direction and strength of statistical association were measured by the odds ratio with 95% CI using multivariable logistic regression analysis. Adjusted odds ratio along with 95% CI was estimated to identify factors associated with BBFS. In this study P-value < 0.05 was considered to declare a result as statistically significant.

### Operational definitions

**Blood/body exposure.** Those exposed to blood, vomits, urine, sputum, saliva, amniotic fluid, exudative fluids [30, 31].

**Personal protective equipment.** Barriers and filters between the worker and the hazard [1].

### Ethical consideration and consent to participate

Ethical clearances were obtained from institutional review boards (IRB) of Addis Ababa University. An official letter of permission was submitted to the Addis Ababa city health bureau. A formal letter was submitted to selected public health institutions from the bureau. The respondents were informed about the objective, purpose, risks, and benefits of the study and the right to refuse to participate, and all subjects provided informed written consent for inclusion before they participated in the study.

The study posed a low or no more than minimal risk to the study participants. Also, the study did not involve any invasive procedures. Moreover, the confidentiality of information was guaranteed by using code numbers rather than personal identifiers and by keeping the data locked.

## Results

### Socio-demographic characteristics

A total of 424 respondents completed the questionnaire yielding a response rate of 97%. The age distribution of the respondents ranges from 20–56 years, with a mean and median age of 30.11 (SD ± 5.37), 26.5 years respectively, and 219 (51.7%) were between the age group of 25–29 years. Nearly two-thirds of participants, 258 (60.8%) were males. Unmarried accounted for 253(59.7%). In this study majority, four hundred and ten (96.7%) participants were holders of degrees and above. More than four out of five earned a monthly salary of ≤8000 Ethiopian birrs. Two-third, 295 (69.6%), had less than five years' work experience (Table 1).

### Behavior, knowledge, and practice of midwives

Out of the respondents, less than half, 191(45%) responded that they had followed universal precautions regularly, and 187 (44.1%) usually use safety deposit boxes to discard sharp materials. Concerning the protection of oneself and the patient from infection, more than three fourth 362 (85.4%) of respondents indicated that they wear gloves appropriately. The wearing of gowns and aprons had responded by 300(70.8%). Among study participants, 195 (46%) had responded that they had job-related stress, and 246 (58%) answered that they were dissatisfied with their job. More than two-thirds of 294 (69.3%) of participants reported that they had always washed their hands immediately after handling contaminated objects while 246 (58.5%) always wash their hands before putting on gloves. Three hundred forty-seven 347

**Table 1. Socio-demographic characteristics of midwives working in government health institutions of Addis Ababa city, Ethiopia, 2020.**

| Variables | Category | Number | % |
|---|---|---|---|
| Age | 20–24 | 35 | 8.3 |
| | 25–29 | 219 | 51.7 |
| | 30–34 | 87 | 20.5 |
| | 35–39 | 55 | 12.9 |
| | ≥40 | 28 | 6.6 |
| Gender | Male | 258 | 60.8 |
| | Female | 166 | 39.2 |
| Marital status | Married | 171 | 40.3 |
| | Single | 253 | 59.7 |
| Educational status | Diploma | 14 | 3.3 |
| | Degree and above | 410 | 96.7 |
| Monthly salary | ≤5000 | 165 | 38.9 |
| | 5001–8000 | 179 | 42.2 |
| | ≥8001 | 80 | 18.9 |
| Year of services | < 5 years | 295 | 69.6 |
| | 5–10 years | 103 | 24.3 |
| | ≥11 | 26 | 6.1 |

**Table 2. Behavior, knowledge, and practice related characteristics of midwives working in public health institutions of Addis Ababa city, Addis Ababa, Ethiopia, 2020.**

| Variables | Categories | Frequency | % |
|---|---|---|---|
| Regularly applying universal precautions | Yes | 191 | 45 |
| | No | 233 | 55 |
| Immediate actions following exposure | Washing with soap and water | 218 | 89.3 |
| | Washing with iodine or alcohol solution | 29 | 11.9 |
| | Got tested for HIV | 96 | 39.3 |
| | Took post-exposure prophylaxis (PEP) | 42 | 17.2 |
| | Took tetanus anti-toxoid (TAT) | 2 | 0.8 |
| Personal & patient protection | Wear gloves appropriately | 362 | 85.4 |
| | Proper handwashing | 301 | 71 |
| | Wearing gowns and aprons | 300 | 70.8 |
| | Wearing of eye goggles, masks, and shields | 276 | 65.1 |
| Job-related stress | Yes | 192 | 46 |
| | No | 232 | 54 |
| Job satisfaction | Satisfied | 176 | 42 |
| | Dissatisfied | 248 | 58 |
| Ever had tested for HIV/AIDS | Yes | 385 | 90.8 |
| | No | 39 | 9.2 |
| Ever had tested HBV or HCV | Yes | 351 | 79.8 |
| | No | 73 | 20.2 |
| Appropriate hand wash before putting on gloves | Always | 150 | 35.4 |
| | Sometimes | 248 | 58.5 |
| | Never | 26 | 6.1 |
| Hand wash after handling contaminated objects | Always | 294 | 69.3 |
| | Sometimes | 125 | 29.5 |
| | Never | 5 | 1.2 |
| Hand wash after contact with blood or mucous membrane | Always | 347 | 81.8 |
| | Sometimes | 69 | 16.3 |
| | Never | 8 | 1.9 |
| Availability of antiseptic hand rub | Always | 142 | 33.5 |
| | Sometimes | 257 | 60.6 |
| | Never | 25 | 5.9 |
| Use protective barriers appropriately | Always | 191 | 46.7 |
| | Sometimes | 223 | 52.6 |
| | Never | 10 | 2.4 |

(81.8%) reported that they washed their hands always after contact with blood or mucous membrane (Table 2).

## Prevalence of blood/body fluid splashes

From a total of 424 midwives who participated in the study occupational exposure to blood/body fluid splash at least once was 198 (46.7%) with 95%CI: 41.3–52.2. Among those exposed to BBFS in the last year, 163(82.3%) of them encountered one to two times, and 14 (7.1%) faced five and more times. From those who were exposed, the status of the source of exposure was unknown for 79 (39.5%), whereas, 25(12.5%) were known HIV positive, and 16 (8%) were clinically suspected HIV positive. However, 77(38.5%) known as HIV negative. Only, 3 (1.5%) of the source of exposure was reported unknown hepatitis B positive. Of those BBFS, 138

(60.8%) exposure was encountered to skin. The splash to the broken membrane of the skin, eyes, and mouth were 16 (7.9%), 51 (25.6%), 11 (5.7%) respectively.

## Predictors of blood/body fluid splash

According to multivariable logistic regression working in two shifts(working for > 12 hours), not received IP and safety practice training, not regularly apply universal precautions, job-related stress, an average monthly salary of 5001–8000 were independent predictors of BBFS.

Health care professionals who did not regularly apply universal precautions were 8 times more likely to have blood/body fluid splash (AOR 8, 95% CI 6.55–14.8) than their counterparts. Participants who did not receive IP and safety practice training were four times more likely to have blood/body fluid splash (AOR 4.27, 95% CI 1.94–9.41) than those who did receive training. The odds ratio was two times higher for those working in two shifts (more than 12 hours) (AOR 2.36, 95% CI 1.034–5.4) than one shift (≤ 12 hours). The odds of exposure to BBFs were two times increased for those who had job-related stress (AOR 2.18, 95% CI 1.07–4.49) than their counterpart. Respondents who get an average monthly salary of 5001–8000 Ethiopian birr were 70% less likely to be exposed to BBFS (AOR 0.305, 95% CI 0.105–0.88) than those who had gained less than 8000 Ethiopian birrs (Table 3).

**Table 3. Predictors of BBFs among midwives working in public health institutions of Addis Ababa city, Ethiopia 2020.**

| Variables | BBFs | | | COR(95% CI) | AOR(95% CI) |
|---|---|---|---|---|---|
| | Yes = 198 | No = 226 | | | |
| Gender | | | | | |
| Male | 163(84.8%) | 95(42%) | | 6.4(4.1–10.1) | 2.1(0.99–4.33) |
| Female | 35(15.2%) | 131(58%) | | 1 | |
| Marital status | | | | | |
| Married | 50(25.3%) | 121(53.5%) | | 1 | |
| Unmarried | 148(74.7%) | 105(46.5%) | | 3.4(2.3–5.2) | 6.02(0.42–10.02) |
| Did you work in shift? | | | | | |
| Yes | 173(87.4%) | 173(76.5%) | | 2.1(1.26–3.6) | 2.36(1.034–5.4)* |
| No | 25(12.6) | 53(23.5%) | | 1 | 1 |
| Work hours per week | | | | | |
| ≥40 hours | 158(79.8%) | 80(35.4%) | | 7.2(4.6–11.2) | 2.01(0.002–9.03) |
| ≤39 hours | 40(20.2%) | 146(64.6%) | | 1 | |
| Receiving IP and safety training in the last year | | | | | |
| Yes | 27(13.6%) | 184(81.4%) | | 1 | 1 |
| No | 171(86.4%) | 42(18.6%) | | 27.7(16.4–46.9) | 4.27(1.94–9.41)*** |
| Use of safety containers regularly | | | | | |
| Yes | 16(8.1%) | 171(75.6%) | | 1 | 1 |
| No | 182(91.9%) | 55(24.4%) | | 35.4(19.5–64.1) | 21.01(0.99–26.02) |
| Regularly applying universal precautions | | | | | |
| Yes | 11(5.6%) | 180(79.6%) | | 1 | 1 |
| No | 187(94.4%) | 46(20.4%) | | 66.5(33.4–132.5) | 8.43(6.55–14.8)*** |
| Job-related stress | | | | | |
| Yes | 151(76.3%) | 44(19.5%) | | 13.3(8.4–21.1) | 2.18(1.07–4.49)* |
| No | 47(23.7%) | 182(80.5%) | | 1 | 1 |
| Year of service | | | | | |
| 1–5 years | 153(77.3%) | 142(62.8%) | | 2.9(1.2–7.2) | 0.23(0.25–2.65) |
| 6–10 years | 38(19.2%) | 65(28.8%) | | 1.6(0.6–4.1) | |

*(Continued)*

**Table 3.** (Continued)

| Variables | BBFs | | COR(95% CI) | AOR(95% CI) |
|---|---|---|---|---|
| | Yes = 198 | No = 226 | | |
| ≥ 10 years | 7(3.5%) | 19(8.4%) | 1 | 1 |
| Monthly salary category | | | | |
| ≤5000 birr | 88(44.4%) | 77(34%) | 1.26(0.74–2.2) | |
| 5001–8000 birr | 72(36.4%) | 107(47.3%) | 0.74(0.437–1.3) | 0.305(0.105–0.88)* |
| ≥8001 birr | 38(19.2%) | 42(18.7%) | 1 | 1 |
| Job satisfaction | | | | |
| Satisfied | 46(23.2%) | 132(58.4%) | 1 | 1 |
| Dis satisfied | 152(76.8%) | 94(41.6%) | 4.6(3–7.1) | 3.59(0.87–6.07) |

*p = 0.04

**p = 0.01

***p = 0.001.

## Discussions

In this study, the prevalence of BBFS at least once in their work-life was 198 (46.7%) with 95% CI: 41.3–52.2. The prevalence was in line with a study done in Australia 42% [20]. But it was significantly higher than some findings in Eastern Ethiopia (20.2% and elsewhere 36.1%) [18, 32]. In contrast, this proportion remains low compared to the study done in Gonder and Tigray Ethiopia at 62.9%, 56.3% respectively [2, 19]. The reason for this difference could be methodological differences and variation in study area and periods, and under-reporting in this study. And also may be due to the difference in and the availability of resources. Thus, the study was done in an urban area the availability of materials to apply safety precautions was better than in rural settings. Work cultures may be significantly affecting the outcome the study included only midwives in whose work area is more susceptible for BBFS.

The most potential determinant factor associated with BBFS was not regularly applying universal precaution (AOR 8.43, 95% CI 6.55–14.8). Those who did not apply universal precautions were 8 times more exposed to BBFS than those who apply. This is consistent with studies done in Mongolia [33] and Northwest Ethiopia [34, 35]. This might be because those who did not adhere to universal precautions were more likely to incur occupational exposure due to they are working unprotected. The reasons not to wear universal precautions were due to lack of materials, absence of training.

In this study, midwives who worked >12 hours were twice more likely to be exposed to BBFS (AOR 2.36, 95% CI 1.034–5.4) than those who worked <12 hours. This finding was supported by the study done in Germany [14], and Ondo state, Nigeria [36]. Since >12 hours' work has a high clinical activity schedule, with full of workload and they will stay more than 12 hours in the workplace, so as time elapses they become tired and negligent to apply universal precautions [37, 38].

The result of the study indicates that lack of training on IP and safety practice in the last year was a significant determinant to BBFS (AOR 4.27, 95% CI 1.94–9.41). This may be because training enhances or contributed to change the knowledge attitude and practice of individuals. Similarly, a study carried out in Debre Birhan Ethiopia and Kenya [39, 40] revealed that the absence of onsite training on IP and safety practice was the main contributor to occupational exposure. Also maybe knowledge and practice acquired during training necessarily translate into practice on prevention measures and safety precautions.

Midwives who had job-related stress were two times more likely to be exposed to BBFS (AOR 2.18, 95% CI 1.07–4.49) than those who did not have job-related stress. This is in agreement with the study done in Northwest Ethiopia [35]. This may be justified since those who had job-related stress could not focus on their jobs and they may be predisposed to BBFS. And also stress may lead to a lack of concentration on work [41, 42].

Respondents who earned a monthly salary of 5000–8000 birr were 70% less likely to be exposed to BBFS (AOR 0.30, 95% CI 0.105–0.88) when compared to respondents who got a monthly salary of ≤5000. Possibly may be due to their satisfaction with their monthly salary, they might be involved seriously in their activities [25, 43, 44]. Also, those who earned ≤5000 might be those with newly deployed to work so they may be less experienced and training could be given first for those senior staffs.

## Conclusion

The study revealed that nearly half of midwives were exposed to BBFS. This highlights the need for key stakeholders such as policymakers and service providers to design appropriate policies to avert this magnitude and making the environment enabling to comply with standard precautions. Also, this study was done in urban health settings, and the institutions are better equipped with personal protective materials than the rural settings. So, we recommend that this study may be done by including rural setting institutions and by including other health professionals that are susceptible to BBFS at work. Formal training on infection prevention and safety practice to apply universal precautions will be needed from the concerned bodies to prevent exposures to blood/body fluid splash.

## Strength and limitation of the study

### Strength of study

This study provides information about the burden of occupational exposure related to BBFs. It is an unsearched/neglected thematic area of the health care system and it adds value for the system and serves as a baseline for researchers.

### Limitation of study

Information was obtained by asking the respondents, so this may incur recall bias. And due to the self-administered questionnaire, it was susceptible to social desirability bias HCPs might report socially acceptable responses than their actual day-to-day practice that may underestimate the prevalence of the outcome.

The study was only urban setting institutions and midwives working in this context. So, it lacks generalizability for midwives working in rural areas. The study was done only on midwives so the study will sound better if it includes other health care professionals.

Because of the cross-sectional study design, the study could not answer the direction of causality between independent factors and outcome.

## Declarations

The content of the study is solely the responsibility of the authors.

## Supporting information

**S1 File. Questionnaire.**
(DOCX)

**S2 File. Minimal data set.**
(SAV)

## Acknowledgments

Special acknowledgment goes to the Addis Ababa University, College of Health Sciences of the School of Public Health for the support provided to undertake this study. Our deepest gratitude goes to the Addis Ababa city health bureau and selected health institutions.

We also express our gratitude to the staff of all selected health institutions who participated in the study.

## Author Contributions

**Conceptualization:** Solomon Shitu, Getachew Adugna.

**Data curation:** Solomon Shitu.

**Formal analysis:** Getachew Adugna.

**Methodology:** Haimanot Abebe.

**Software:** Haimanot Abebe.

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
