## [Decision Letter · Decision Letter 0]

5 Mar 2021

PONE-D-20-40344

Occupational Exposure to Blood/ Body Fluid Splash and Its Predictors among Midwives Working in Public Health Institutions at Addis Ababa City Ethiopia, 2020

PLOS ONE

Dear Mr Ayen,

Thank you for submitting your manuscript to PLOS ONE. After careful consideration, we feel that it has merit but does not fully meet PLOS ONE’s publication criteria as it currently stands. Therefore, we invite you to submit a revised version of the manuscript that addresses the points raised during the review process.

We look forward to receiving your revised manuscript.

Kind regards,

Yibeltal Assefa Alemu, M.D., M.Sc., Ph.D.

Academic Editor

PLOS ONE

Journal Requirements:

2.We suggest you thoroughly copyedit your manuscript for language usage, spelling, and grammar. If you do not know anyone who can help you do this, you may wish to consider employing a professional scientific editing service.  

Reviewers' comments:

Reviewer's Responses to Questions

**Comments to the Author**

1. Is the manuscript technically sound, and do the data support the conclusions?

Reviewer #1: Partly

Reviewer #2: Partly

2. Has the statistical analysis been performed appropriately and rigorously? 

Reviewer #1: Yes

Reviewer #2: Yes

3. Have the authors made all data underlying the findings in their manuscript fully available?

Reviewer #1: Yes

Reviewer #2: Yes

4. Is the manuscript presented in an intelligible fashion and written in standard English?

Reviewer #1: No

Reviewer #2: No

5. Review Comments to the Author

Reviewer #1: Thank you for the opportunity to review this manuscript. This is an important piece of work with relevance to practice. I have added comments and suggestion on the attached PDF but generally feel the paper would benefit from a thorough edit. The background needs a major rewrite to clearly provide background to the study and provide evidence of what is known about BBFS in Ethiopia and why Midwives are the target population. The discussion and concllusion laos need careful consideration and linking to exisiting literature to formualte stronger recommendation to practice, policy and future research.

Reviewer #2: The authors need to address some important comments before this paper can be considered for publication.

Introduction

I would like to suggest the authors include a justification why they are interested only in midwives and how the risk of exposure to blood/body fluid splash is varied across health professionals supported with prevalence figures. In the context of the Ethiopian health system, all health facilities have no midwives. Therefore, still, other health professionals such as nurses/public health officers/doctors are at risk for exposure.

Methods

Study setting – the risk of exposure could be high in health professionals working in rural settings than in health professionals working in urban. It would be great if the authors could consider the effect of health facilities setting.

Source Population – does the result will be generalized only to those midwives working in AA? “All midwives working in public health institutions of Addis Ababa city’’ – is the population where your sample was drawn i.e., study population unless the authors' hypothesis was only for urban midwives.

Inclusion criteria – is there unregistered midwives working in the current Ethiopian health system? I would suggest the authors modify this.

Sample size determination – I would recommend the authors to include the sample size assumptions briefly such as proportion from the previous study, margin of error, etc.

Sampling procedures – “…. simple random sampling technique until the required number is obtained.” For simple random sampling techniques, all eligible study participants should be identified before the data collection starts. Including study participants, until the required number is obtained will work for systematic sampling techniques.

Exposure to blood/body fluid splash – (Yes/No)

Variables

The independent variables except the demographic variables need to be clearly defined.

Statistical analysis

I would like to ask the authors used variables with the (P-value < 0.25) to include in the multivariable analysis. For the authors' reference – to read applied logistic regression book by Hosmer and Lemeshow.

Result

Socio-demographic characteristics – report median age instead of the mean. The min and max age showed the distribution is not normal.

Regression output – interpret only the multivariable results.

Discussion

This section is very weak and needs to be improved.

6. PLOS authors have the option to publish the peer review history of their article (what does this mean?). If published, this will include your full peer review and any attached files.

Reviewer #1: No

Reviewer #2: **Yes: **Yalemzewod Gelaw (PhD)

---

## [Author Response · Author response to Decision Letter 0]

10 Mar 2021

Author’s Point-by-Point Response to the Reviewer's and Editors Reports

Title: Occupational Exposure to Blood/ Body Fluid Splash and Its Predictors among Midwives Working in Public Health Institutions at Addis Ababa City Ethiopia, 2020 

Institution-based cross-sectional study

Corresponding author: Solomon Shitu Ayen (solomonsht7@gmail.com)

Authors

Getachew Adugna (getchadu2013@yahoo.com)

Haimanot Abebe (haimanotabebe78@gmail.com)

ID: PONE-D-20-40344

Journal: PLOS ONE

 Article type: Research article

First of all, the authors would like to thank PLOS ONE Journal editors and the respective reviewers for reviewing our manuscript and providing the necessary comments to be corrected. As per the comments given, we have made corrections point by point to comment. The authors tried to answer all the issues raised by the editorial team and reviewers. Please note that we gave our response in blue font color.

Point by point response to editor

Comment “Please ensure that your manuscript meets PLOS ONE's style requirements, including those for file naming”. And we suggest you thoroughly copyedit your manuscript for language usage, spelling, and grammar

Response 1: Thanks for your great suggestion and timely comments. Corrected 

Point by point response to Reviewer# 1

Dear, 

Question1; Grammatical edits within the document

Response 1: Thanks for your great suggestion and timely comments. Corrected grammatical errors throughout the document

Question2; Why target this group of health care professionals - not clear from the background

Response 2: Thanks for your great suggestion and timely comments. We included only midwives was as in general little is known about blood and body fluid splash among all health professionals and midwives working more focus on conducting delivery in which they are more susceptible for fluids and blood splash during attending delivery. Because they are nearest to exposure like amniotic fluid, meconium, blood, vomiting, and so on. 

Question3; Rewrite the conclusion 

Response 3: Thanks for your great suggestion and timely comments. Corrected as, “The study revealed that nearly half of midwives exposed to BBFS. This sparks light to concerned bodies like policymakers and stack holders to design appropriate policy to avert this magnitude and making the environment enabling to comply with standard precautions. Also, this study was done in urban health settings, and the institutions are better equipped with personal protective materials than the rural settings. So, we recommend that this study may be done by including rural setting institutions and by including other health professionals that are susceptible to BBFS at work. Formal training on infection prevention and safety practice to apply universal precautions will be needed from the concerned bodies to prevent exposures to blood/body fluid splash.”

Question 4; Rewrite the background

Response 4: Thanks for your great suggestion and timely comments. Corrected as Blood/body fluid splash (BBFS) are hazards to health care workers in their working area. Around twenty bloodborne pathogens are known to transmit through this occupational exposure [1]. This problem alters the health status of health care professionals (HCPs) in different ways, including physically, mentally, and psychologically [2]. 

No occupation or work is free from health hazards. People are always under a certain amount of health risk wherever they are working [3]. Health care professionals are exposed to a wide range of occupational health and safety hazards [4]. An estimated 20 million work-related injuries and 390, 000 new work-related illnesses occur each year in the United States and China 66.3% of HCPs per year exposed to BBFS; however, the number of occupational diseases and injuries reported each year is much more [4, 5]. 

Nowadays, workplace injuries, especially BBFS prevalence, are increasing among health workers [6, 7]. Occupational exposure to BBFS in healthcare facilities constitutes a significant risk of transmission of human immune deficiency virus (HIV) and other blood-borne pathogens to healthcare workers [8-10]

As estimated by the Center for Disease Control (CDC) that every year more than three million HCPs are exposed to BBFs, through sharps, plus mucocutaneous injuries in the United States [10]. Due to BBFS, the possibility of infections ranges from as low as 0.2–0.5% for HIV to as huge as 3–10% for Hepatitis C virus (HCV) and 40% for Hepatitis B virus (HBV) [11]. World health organization (WHO) estimates that nearly 3 million HCPs experience occupational exposure to blood-borne viruses HBV, HCV, and HIV has been reported to be 2.1 million, 926,000, and 327,000 respectively [12-15]. From this 90% of this exposure to infection was resulted from low resource countries [14]. In Ethiopia, studies indicated that the prevalence of HCV due to BBFS exposure was 0.9- 5.8% [3, 16].

Health care workers in Africa experience from two to four needle-stick injuries per year on average, with Nigeria, Tanzania, and South Africa reporting on average of 2.10 injuries per HCPs per year [17]. 

Even though there is no national data regarding the prevalence of BBFS among midwives in Ethiopia previous studies in Eastern, Southern, and Northern parts have shown the increased risk of occupational blood exposure [17-19]. The majority of these studies were only confined to a single institution and workplace [18, 19]. The high rate of HCPs is affected by the exposure [2]. Despite the presence of many possibilities to reduce the burden of occupational exposure to BBFS, it is not yet recognized. Hence this research was carried out on midwives working at public health institutions of Addis Ababa city delivery unit due to they are more exposed and the first line for BBFS. 

Question 5; why you focus only on midwives?

Response 5: Thanks for your great suggestion and timely comments. Even though all health professionals are exposed to BBFS, midwives are the first line for fluid and blood exposure. Because obstetrics is a bloody profession and they are the nearby professional. And all health institutions have employed midwives in Addis Ababa city.

Question 6; Study Population 

Randomly selected midwives working in public health institutions of Addis Ababa city.

“How was this done?”

Response 6: Thanks for your great suggestion and timely comments. By using a systematic random sampling technique 

Question 7; On sampling procedure, we said The registers of all midwives were obtained from the administration of the offices. “Obtaining this list of names does pose some ethical considerations. Need to clarify how the researchers obtained access to this register. Had the midwives provided informed consent for their names and contact details to be released for research purposes?”

Response 7: Thanks for your great suggestion and timely comments. Even though the study has some ethical issues we directly requested the office by letter but because the study has minimal or no harm on them. 

Question 8; more detail of process needed on sampling procedure

Response 8: Thanks for your great suggestion and timely comments. Possible correction taken by each institution has their number of allocated midwives based on that we classify proportionally based on total sample size. “Six public hospitals from 12 hospitals and 66 health centers from 112 health centers were selected by simple random sampling technique. The total number of midwives working in Addis Ababa city was 2432 and the number of midwives working in selected public health institutions was 1296. To select study participants from selected health institutions the proportional allocation method (each institution's number of midwives was allocated by proportion) was used for each institution. Study participants who fulfilled the inclusion criteria were selected by systematic random sampling technique until the required number is obtained with a kth value of 3.”

Question 9; Data quality control and collection methods 

Using an English version questionnaire “is this a limitation? English literacy levels etc”

Response 9: Thanks for your great suggestion and timely comments. This study was related to their job and doesn’t need that much fluent speaker or writer of English. The interviewees were minimum of diploma certificate so they can understand the questionnaire.

Question 10; in result part “is this reflective of the midwifery workforce demographic in Ethiopia? If not then why more males completed”

Response 10: Thanks for your great suggestion and timely comments. Your concern was correct midwifery profession was perceived as a profession of women in the previous era but in the current situation this is changed and more males are involved as midwife professionals. That may be the reason that in this study the percentage of males was greater than females. 

Question 11; These data are not presented in a table - sentence structure needs review to make results clear. Also review terminology around HIV/AIDS - was 12.5 living with HIV and been diagnosed with an AIDS-defining illness or is it more accurate to say 12.5% were HIV positive or known to be living with HIV.

Response 11: Thanks for your great suggestion and timely comments. Corrected 

Question 12; the meaning of working in shifts is unclear - please clarify

Response 12: Thanks for your great suggestion and timely comments. Corrected as working in two shifts >12 hours work or day and night is more likely exposed than working in only one shift (working for ≤12 hours)

Point by point response to Reviewer# 2

Dear, Yalemzewod Gelaw (PhD) 

Question1; I would like to suggest the authors include a justification why they are interested only in midwives and how the risk of exposure to blood/body fluid splash is varied across health professionals supported with prevalence figures. In the context of the Ethiopian health system, all health facilities have no midwives. Therefore, still, other health professionals such as nurses/public health officers/doctors are at risk for exposure.

Response 1: Thanks for your great suggestion and timely comments. We included only midwives was as in general little is known about blood and body fluid splash among all health professionals and midwives working more focus on conducting delivery in which they are more susceptible for fluids and blood splash during attending delivery. Because they are nearest to exposure like amniotic fluid, meconium, blood, vomiting, and so on. Since the study was done in Addis Ababa city the work of the delivery unit was covered by midwives in all health institutions even if the number of professionals is not equal to the standard. 

Question 2; “Study setting – the risk of exposure could be high in health professionals working in rural settings than in health professionals working in urban. It would be great if the authors could consider the effect of health facilities setting”.

Response 2: Thanks for your great suggestion and timely comments. This is a correct issue that the study was better sound than this if it included the rural setting but due to different reasons, it was done only in the urban area even though little was known on the topic. We recommend other researchers incorporate rural settings in future studies. 

Question 3; Source Population – does the result will be generalized only to those midwives working in AA? “All midwives working in public health institutions of Addis Ababa city’’ – is the population where your sample was drawn i.e., study population unless the authors' hypothesis was only for urban midwives.

Response 3: Thanks for your great suggestion and timely comments. Yes, this study was generalized for urban setting midwives. Because the source population was midwives working in public health institutions of Addis Ababa city 

Question 4; Inclusion criteria – is there unregistered midwives working in the current Ethiopian health system? I would suggest the authors modify this.

Response 4: Thanks for your great suggestion and timely comments. Corrected as “All midwives working in Addis Ababa city”

Question 5; Sample size determination – I would recommend the authors to include the sample size assumptions briefly such as proportion from the previous study, margin of error, etc.

Response 5: Thanks for your great suggestion and timely comments. Corrected as “A separate sample size was calculated for each specific objective by using both single and double population proportion formula was used. For the first specific objective, (To determine the prevalence of occupational exposure to BBFS among midwives working in Addis Ababa city) single population proportion was used. 

n= (Z α/2)2 P (1-P) by the assumption of n= minimum sample size required for the study

 d2 

 (Z α/2)2 = standard normal distribution with 95% confidence interval, P= 21% (prevalence of occupational exposure and behavior of health care workers in Ethiopia) [18], d= is a tolerable margin of error (d=0.04) 

The sample size of the first objective was greater than that of the second objective. By adding a 10% non-response rate from the largest sample size the calculated sample size was 438.”

Question 6; Sampling procedures – “…. simple random sampling technique until the required number is obtained.” For simple random sampling techniques, all eligible study participants should be identified before the data collection starts. Including study participants, until the required number is obtained will work for systematic sampling techniques.

Response 6: Thanks for your great suggestion and timely comments. Corrected as “The total number of midwives working in Addis Ababa city public health institutions was 2432 and number of midwives working in selected institutions was 1296. To select study participants from selected health institutions the proportional allocation method (each institution's number of midwives was allocated by proportion) was used. Study participants who fulfilled the inclusion criteria were selected by systematic random sampling technique until the required number is obtained with a kth value of 3”.

Question 7; The independent variables except the demographic variables need to be clearly defined.

Response 7: Thanks for your great suggestion and timely comments. Corrected as Work-related factors: (Training on IP, working environment, year of service, educational level)

Behavioral factors: (Job satisfaction, reporting after the exposure, using the universal precautions and personal protective equipment)

Question 8; I would like to ask the authors to use variables with the (P-value < 0.25) to include in the multivariable analysis. For the authors' reference – to read applied logistic regression book by Hosmer and Lemeshow.

Response 8: Thanks for your great suggestion and timely comments. Corrected as “The collected data were checked and reviewed for completeness, coded, and entered into Epi data version 3.1 and exported to SPSS version 24 for analysis. Descriptive statistics were used to determine the frequency of different variables. Association between dependent and independent variables were examined using bi-variable and multivariable logistic regression models. Those variables which were (P-value < 0.25) [18] in the bivariate analysis were included in the final model of multivariate analysis to control all possible confounders. Multi-collinearity was checked to see the linear correlation among the independent variables by using standard error. Variables with a standard error of > 2 were dropped from the multivariable analysis. Model fitness was checked with the Hosmer-Lemeshow test. The direction and strength of statistical association were measured by the odds ratio with 95 % CI using multivariable logistic regression analysis. Adjusted odds ratio along with 95% CI was estimated to identify factors associated with BBFS. In this study P-value < 0.05 was considered to declare a result as statistically significant”.

Question 9; Socio-demographic characteristics – report median age instead of the mean. The min and max age showed the distribution is not normal.

Response 9: Thanks for your great suggestion and timely comments. Corrected as “The age distribution of the respondents ranges from 20-56 years, with a mean and median age of 30.11 (SD ± 5.37), 26.5 years respectively and 219 (51.7%) were between the age group of 25-29 years”.

Question 10; Regression output – interpret only the multivariable results. 

Response 10: Thanks for your great suggestion and timely comments. Corrected as “In multivariable logistic regression working in >12 hours shift, not receive IP and safety practice training, not regularly apply universal precautions, job-related stress, an average monthly salary of 5001-8000 were found independent predictors of BBFS.

Health care professionals who did not regularly apply universal precautions were 8 times more likely to have blood/body fluid splash (AOR 8, 95% CI 6.55 - 14.8) than their counterparts. Participants who did not receive IP and safety practice training were four times more likely to have blood/body fluid splash (AOR 4.27, 95% CI 1.94 - 9.41) than those who did receive training. The odds ratio was two times higher for those working in two shifts (more than 12 hours) (AOR 2.36, 95% CI 1.034-5.4) than one shift (≤ 12 hours). The odds of exposure to BBFs were two times increased for those who had job-related stress (AOR 2.18, 95% CI 1.07-4.49) than their counterpart. Respondents who get an average monthly salary of 5001-8000 Ethiopian birr were 70% less likely to be exposed to BBFS (AOR 0.305, 95% CI 0.105 - 0.88) than those who had gained less than 8000 Ethiopian birrs.

Question 11; Discussion section is very weak and needs to be improved.

Response 11: Thanks for your great suggestion and timely comments. Rewritten and improved as much as possible

---

## [Decision Letter · Decision Letter 1]

14 Apr 2021

PONE-D-20-40344R1

Occupational Exposure to Blood/ Body Fluid Splash and Its Predictors among Midwives Working in Public Health Institutions at Addis Ababa City Ethiopia, 2020

PLOS ONE

Dear Solomon,

Thank you for submitting your manuscript to PLOS ONE. After careful consideration, we feel that it has merit but does not fully meet PLOS ONE’s publication criteria as it currently stands. Therefore, we invite you to submit a revised version of the manuscript that addresses the points raised during the review process. We recommend that you carefully consider the comments from reviewer 1 in your revision. 

Please submit your revised manuscript by 28 May 2021. If you will need more time than this to complete your revisions, please reply to this message or contact the journal office at plosone@plos.org. Please include the following items when submitting your revised manuscript:

We look forward to receiving your revised manuscript.

Kind regards,

Yibeltal Alemu, M.D., M.Sc., Ph.D.

Academic Editor

PLOS ONE

Journal Requirements:

Reviewers' comments:

Reviewer's Responses to Questions

**Comments to the Author**

1. If the authors have adequately addressed your comments raised in a previous round of review and you feel that this manuscript is now acceptable for publication, you may indicate that here to bypass the “Comments to the Author” section, enter your conflict of interest statement in the “Confidential to Editor” section, and submit your "Accept" recommendation.

Reviewer #1: (No Response)

Reviewer #2: All comments have been addressed

2. Is the manuscript technically sound, and do the data support the conclusions?

Reviewer #1: Yes

Reviewer #2: Yes

3. Has the statistical analysis been performed appropriately and rigorously? 

Reviewer #1: Yes

Reviewer #2: No

4. Have the authors made all data underlying the findings in their manuscript fully available?

Reviewer #1: Yes

Reviewer #2: Yes

5. Is the manuscript presented in an intelligible fashion and written in standard English?

Reviewer #1: No

Reviewer #2: No

6. Review Comments to the Author

Reviewer #1: Thank you for the opportunity to review your revised manuscript. This is an important piece of work with relevance to practice. The authors have addressed the reviewer’s comments, but the paper still requires a thorough edit. This is particularly needed in the newly added sections.

Some of the information added to the responses to the reviewer’s queries needs to be added to the actual manuscript to strengthen the paper – see attached

The discussion needs to also increase the discussion about why midwives are at increased risk and if your findings are relevant to any other health care workers placed in high risk areas such as emergency / trauma setting and ?

Reviewer #2: The authors have addressed most of my comments and they have made significant change. However, the manuscript still needs standard language edition before publication.

7. PLOS authors have the option to publish the peer review history of their article (what does this mean?). If published, this will include your full peer review and any attached files.

Reviewer #1: No

Reviewer #2: **Yes: **Yalemzewod Gelaw

---

## [Author Response · Author response to Decision Letter 1]

22 Apr 2021

Author’s Point-by-Point Response to the Reviewer's and Editors Reports

Title: Occupational Exposure to Blood/ Body Fluid Splash and Its Predictors among Midwives Working in Public Health Institutions at Addis Ababa City Ethiopia, 2020 

Institution-based cross-sectional study

Corresponding author: Solomon Shitu Ayen (solomonsht7@gmail.com)

Authors

Getachew Adugna (getchadu2013@yahoo.com)

Haimanot Abebe (haimanotabebe78@gmail.com)

ID: PONE-D-20-40344

Journal: PLOS ONE

 Article type: Research article

First of all, the authors would like to thank PLOS ONE Journal editors and the respective reviewers for reviewing our manuscript and providing the necessary comments to be corrected. As per the comments given, we have made corrections point by point to comment. The authors tried to answer all the issues raised by the editorial team and reviewers. Please note that we gave our response in blue font color.

Point by point response to editor

Comment “We recommend that you carefully consider the comments from reviewer 1 in your revision”. 

Response 1: Thanks for your great suggestion and timely comments. Corrected 

Point by point response to Reviewer# 1

Dear, 

Question1; comment on conclusion

Response 1: Thanks for your great suggestion and timely comments. Corrected as you recommendation

Question 2; the paper still requires a thorough edit of language 

Response 2: Thanks for your great suggestion and timely comments. We have taken corrections concerning to any English errors and typos by the help of a man who have PhD in English language. Furthermore, we have also taken correction for any English errors using online grammar and English typo correctors apps (We used the following links- 

https://app.grammarly.com/?network=g&utm_source=google&matchtype=e&gclid=Cj0KCQjwo-aCBhC-ARIsAAkNQiuJ49UHhl6ibhQfzq9D4wGrbSOeZPv49UoRqSnd4ThQ-KKrPp uBp4aAjLgEALw_wcB&placement=&q=brand&utm_content=486649398671&gclsrc=aw.ds&utm_campaign=brand_f1&utm_medium=cpc&utm_term=grammarly and https://pubsure.researcher.life/author/?active_tab=recent_plan

So, now we have solved all iniquities related with the English language including the tense used and unnecessary capitalization and other typos/ errors

Question 3; relevance to paper - Introduction needs further refining to more clearly link t rationale for why Midwives are considered high risk - group. More information is added to the responses to the reviewer’s queries that could strengthen the paper if inserted into the text body

Response 3: Thanks for your great suggestion and timely comments. Corrected as you recommendation

Question 4; permission for what ? Is this authoriastion to access the register of midwives. This needs to be more specific and clear that this is in line with ethical standards of practice as this sharing of personal details of registered staff would not be allowed in many context under Pravacy acts. .

Response 4: Thanks for your great suggestion and timely comments. The formal letter submitted to the institutions was used as to ask permission and to show the legal step that we go while conducting this study.

Question 5; what is the difference between written and signed

Response 5: Thanks for your great suggestion and timely comments. Written and signed means we or data collectors prepare written document that the participants sign on it.

Corrected as “An official letter of permission was submitted to the Addis Ababa city health bureau. A formal letter was submitted to selected public health institutions from the bureau. The respondents were informed about the objective, purpose, risks, and benefits of the study and the right to refuse to participate, and all subjects provided informed written consent for inclusion before they participated in the study”.

Question 6; The status of the source of exposure was unknown for 79 (39.5%), whereas, 25(12.5%) were known HIV positive, 16 (8%) were clinically suspected HIV positive, however, 77(38.5%) known HIV negative. “sentence structure needs review”

Response 6: Thanks for your great suggestion and timely comments. Corrected as “From those who were exposed, the status of the source of exposure was unknown for 79 (39.5%), whereas, 25(12.5%) were known HIV positive and 16 (8%) were clinically suspected HIV positive. However, 77(38.5%) known HIV negative.”

Question 7; discussion “sentence structure - make prevalence of what clear”

Response 7: Thanks for your great suggestion and timely comments. Corrected as “In this study, the prevalence of BBFS at least once in their work life was 198 (46.7%) with 95%CI: 41.3-52.2.”

Question 8; discussion “was this Australian study involving midwives or who. Same for the other studies cited - need to make sure you are comparing similar groups and if not then discuss what this means”

Response 8: Thanks for your great suggestion and timely comments. There is no study done only on midwives but we used literatures done on BBFS. So we discussed with those literatures done on BBFS.

Question 9; Do you mean midwives who worked across two shifts that is > 12 hours - for the international reader you need to make clear what normal shift working hours are and whether these long hours are the norm so therefore a common risk for midwives. I think it is clearer if you just acknowledge the <12 and >12 hours and not focus on the shifts 

Response 9: Thanks for your great suggestion and timely comments. Corrected 

Question 10; limitation And the study has done only on midwives so better to include other health professionals “sentence structure”

Response 10: Thanks for your great suggestion and timely comments. Corrected as “The study was done only on midwives so the study will sound better if it includes other health care professionals.”

---

## [Editor Report · Decision Letter 2]

4 May 2021

Occupational exposure to blood/ body fluid splash and its predictors among midwives working in public health institutions at Addis Ababa city Ethiopia, 2020

PONE-D-20-40344R2

Dear Solomon Shitu Ayen,

We’re pleased to inform you that your manuscript has been judged scientifically suitable for publication and will be formally accepted for publication once it meets all outstanding technical requirements.

Kind regards,

Yibeltal Alemu, M.D., M.Sc., Ph.D.

Academic Editor

PLOS ONE

---

## [Editor Report · Acceptance letter]

19 May 2021

PONE-D-20-40344R2 

Occupational exposure to blood/ body fluid splash and its predictors among midwives working in public health institutions at Addis Ababa city Ethiopia, 2020
Institution-based cross-sectional study 

Dear Dr. Shitu:

I'm pleased to inform you that your manuscript has been deemed suitable for publication in PLOS ONE. Congratulations! Your manuscript is now with our production department. 

Kind regards, 

on behalf of

Dr. Yibeltal Alemu 

Academic Editor

PLOS ONE